# Pre-analytical handling conditions and protein marker recovery from urine extracellular vesicles for bladder cancer diagnosis

Jisu Lee[1], Eunha Kim[1], Joohee Park[1], Seokjoo Choi[1], Myung-Shin Lee[1,2]*, Jinsung Park[3]*

**1** Department of Microbiology and Immunology, Eulji University School of Medicine, Daejeon, Republic of Korea, **2** Eulji Biomedical Science Research Institute, Eulji University School of Medicine, Daejeon, Republic of Korea, **3** Department of Urology, Uijeongbu Eulji Medical Center, Eulji University School of Medicine, Uijeongbu-si, Republic of Korea

* mslee@eulji.ac.kr (MSL); jspark.uro@gmail.com (JP)

## Abstract

Extracellular vesicles (EVs) contain a variety of biomolecules and provide information about the cells that produce them. EVs from cancer cells found in urine can be used as biomarkers to detect cancer, enabling early diagnosis and treatment. The potential of alpha-2-macro-globulin (A2M) and clusterin (CLU) as novel diagnostic urinary EV (uEV) biomarkers for bladder cancer (BC) was demonstrated previously. To validate the diagnostic value of these proteins in uEVs in a large BC cohort, urine handling conditions before uEV isolation should be optimized during sample transportation from medical centers. In this study, we analyzed the uEV protein quantity, EV particle number, and uEV-A2M/CLU after urine storage at 20°C and 4°C for 0–6 days, each. A2M and CLU levels in uEVs were relatively stable when stored at 4°C for a maximum of three days and at 20°C for up to 24 h, with minimal impact on analysis results. Interestingly, pre-processing to remove debris and cells by centrifugation and filtration of urine did not show any beneficial effects on the preservation of protein bio-markers of uEVs during storage. Here, the importance of optimizing shipping conditions to minimize the impact of pre-analytical handling on the uEVs protein biomarkers was empha-sized. These findings provide insights for the development of clinical protocols that use uEVs for diagnostic purposes.

## Introduction

Extracellular vesicles (EVs) are lipid bilayer secretory vesicles found in body fluids, such as blood, urine, saliva, hydrothorax, cerebrospinal fluid, and breast milk [1, 2]. EVs are secreted by almost all cell types and contain various proteins, lipids, mRNAs, noncoding RNAs, and other metabolites [3, 4]. Urine, the second most used diagnostic biofluid after blood, owing to its non-invasive sample access, availability in large quantity, and easy repeat measurements, contains a mixture of EVs derived from the urogenital tract, including kidneys, bladder, and prostate [5, 6]. Over the last decade, urinary EV (uEV) has emerged as a promising diagnostic

**Funding:** This work was supported by the National Research Foundation of Korea (NRF) grant funded by the Korean government (No. NRF-2022R1F1A1065578 to J.P. and NRF-2022R1I1A3065538 to M-S.L.). The funders had no role in study design, data collection and analysis, decision to publish, or preparation of the manuscript.

**Competing interests:** The authors have declared that no competing interests exist.

tool for various diseases, reflecting physiological and pathological conditions in the kidney, urothelial, and prostate tissues [7–9]. High-throughput proteomics technologies combined with standard analytical methods have identified numerous potential uEV biomarkers for prostate, bladder, and kidney cancers [10–12]; however, many newly discovered candidates have yet to undergo validation in extensive, multi-centered cohort studies, and the clinical transition is yet to be made. Additionally, there is limited research addressing the collection, processing, storage, and delivery of urine, specifically for uEV experiments. Since differences in these pre-analytical variables can confound study data with variable inclusion of EV sub-populations and other contaminants, such as cells, protein aggregates, and uromodulin networks, it is preemptive that standardization of such variables is achieved across research teams. Urine sample transportation conditions are particularly important in large, multicenter studies to discover reliable novel biomarkers.

Previous studies have assessed the effect of urine storage temperature and time on overall miRNA yield [13], protein concentration [14], EV concentration and proteome [15], and nanoparticle concentration [16], with storage time ranging from a week to a year and temperature ranging from room temperature (RT) to −80˚C. Moreover, a study investigated the effect of the storage format (urine or isolated uEV), storage temperature (−20˚C vs. −80˚C), and storage time of up to four years on uEV quality by nanoparticle tracking analysis, electron microscopy, western blotting, and qPCR [17]. A previous report from the Urine Task Force of the International Society for Extracellular Vesicles recommended that although it may be locally required to maintain a low temperature for urine storage, samples, in general, should be stored maximum 8 h before processing and kept at a temperature of maximum 4˚C to avoid bacterial growth, cell lysis, molecular degradation of RNA and protein, and formation of sediments [5]. However, to date, limited research has explored the impact of temperature or duration during the transportation of the samples on the quality of specific proteins associated with individual uEV biomarkers.

Our previous study demonstrated the potential of alpha-2-macroglobulin (A2M) and clusterin (CLU) as novel diagnostic uEV biomarkers for bladder cancer (BC) through a comparative analysis of uEVs acquired from patients with BC before and after surgery using an antibody array [18]. Enzyme-linked immunosorbent assay (ELISA) of uEVs obtained from patients with BC (n = 60) revealed significant upregulation of A2M expression compared with those from the non-cancer group (n = 23). As subsequent large-scale validation studies are necessary to confirm the diagnostic value of A2M, urine samples should be collected from multiple centers, and urine sample handling conditions must be optimized and standardized. To the best of our knowledge, no prior studies have investigated the effects of temperature and duration of sample transportation on the expression of specific protein markers of uEVs. The aim of this study was to determine the optimal handling conditions necessary to maintain the stability of uEV protein markers during sample transportation. Urine samples collected from five patients with BC were compared for the potentially variable expression of two uEV protein markers, including A2M and CLU, in the EVs isolated from each urine sample after storage at 20˚C and 4˚C for 0–6 days, respectively.

## Materials and methods

### Human samples and data collection

Urine samples were obtained from the study participants at the Uijeongbu Eulji Medical Center, Eulji University (Uijeongbu-si, Republic of Korea), between November 2022 and December 2022. First-morning specimens from patients with BC were obtained before transurethral resection of the bladder tumor to analyze the stability of the uEV protein markers A2M and

CLU. BC was staged based on the 2010 TNM staging system [19] and graded according to the 2004 World Health Organization grading system [20]. Procedures involving urine sample collection and analysis for this study were approved by the Institutional Review Board of Uijeongbu Eulji Medical Center, Eulji University (Validation of diagnostic value of unrinary extracellular vesicles-derived protein marker, alpha 2-macroglobulin, for bladder cancer and functional analysis, No. 2022-07-015-001) and conducted according to the principles outlined in the Declaration of Helsinki. All participants were informed of the purpose of the experiment and provided written informed consent before participating.

## Urine processing, storage, and separation of EVs

After urine collection, each sample was divided into two equal volumes and subjected to two handling conditions: with and without pre-processing. The term *pre-processing* refers to a two-step process of centrifugation at 2,000 ×$g$ for 10 min, followed by filtration through a 0.45 μm syringe filter to eliminate cells and debris to generate a cell-free urine sample. uEVs were isolated from a fraction of pre-processed urine samples and stored at –70˚C as a positive control. To alternate urine handling circumstances, urine samples with or without pre-processing were aliquoted into seven tubes and stored at 20˚C or 4˚C. One aliquot per day for one week was transferred and stored at –70˚C. After six days, all stored samples were centrifuged at 2,000 ×$g$ for 10 min, and the supernatant was filtered through a 0.22 μm syringe filter. To isolate uEVs using Exodisc (LabSpinner, Inc., Ulsan, South Korea) for priming, phosphate-buffered saline (PBS) was added to the filter chamber, and the solution was centrifuged using ExoDiscovery (LabSpinner, Inc.) for 5 min to activate the filter. The prepared urine samples were transferred to filter chambers and centrifuged for 10 min to separate the EVs for enrichment. Finally, the collected EVs were washed twice by adding PBS to the filter chambers, and the solution was centrifuged using ExoDiscovery.

## Nanoparticle tracking analysis

The number and size distribution of microparticles in the EV preparations were analyzed using the nanoparticle tracking analyzer ZetaView (Particle Metrix GmbH, Meerbusch, Germany) as described previously [21, 22]. EV preparations were diluted in distilled water and passed through 0.22 μm filters before analysis. The analysis parameters were as follows: maximum area of 1,000; minimum area, 10; minimum brightness, 20; sensitivity,76; shutter, 100; and temperature, 25˚C.

## Cryo-transmission electron microscopy for EVs

Three microliters of the EV sample were placed on a Lacey carbon grid with a mesh size of 200 (Electron Microscopy Sciences, Hatfield, PA). After a 30-second waiting period, excess liquid was blotted, and the grid was submerged in liquid ethane for 4 seconds. Subsequently, the grid was transferred to liquid nitrogen and mounted on a Gatan 4022 cryo-holder to maintain a temperature below -180˚C during cryo-TEM. The TEM images were captured using a Gatan Rio 16 camera.

## Protein-based EV quantification and western blot analysis

Protein-based quantification of the isolated EVs was performed using a micro bicinchoninic acid (BCA) assay kit (Thermo Fisher Scientific, Waltham, MA) according to the manufacturer's instructions. Subsequently, equal volumes of EVs isolated were denatured using 5× sample buffer without dithiothreitol at 95˚C for 10 min and then resolved on a 12% SDS-acrylamide

gel via electrophoresis. The resolved proteins were transferred onto nitrocellulose membranes (GE Healthcare, Solingen, Germany), which were then blocked by incubation in 5% skim milk with 0.1% Tween-20 buffer to minimize the non-specific binding of antibodies. Blocked membranes were then incubated with primary antibodies overnight at 4°C, washed three times with 1× Tris-buffered saline with 0.1% Tween-20 (TBST) buffer, and incubated with horseradish peroxidase (HRP)-conjugated secondary antibody for 1 h at RT. Unbound antibodies were removed by washing with 1× TBST buffer, and immunolabeled proteins were visualized using the Clarity Western ECL Substrate (Bio-Rad, Hercules, CA) and the Amersham ImageQuant 800 system (GE Healthcare). Mouse monoclonal anti-CD63 (sc-5275; Santa Cruz Biotechnology, Santa Cruz, CA), rabbit monoclonal anti-CD9 (ab236630; Abcam, Waltham, MA), HRP-conjugated goat anti-mouse IgG antibody (A90-116P; Bethyl Laboratories, Montgomery, TX), and HRP-conjugated goat anti-rabbit IgG antibody (A120-101P; Bethyl Laboratories) were used.

### ELISA for A2M and CLU

The levels of A2M and CLU in uEVs were analyzed using the human A2M DuoSet ELISA kit (R&D Systems) and human Clusrein DuoSet ELISA kit (R&D Systems), respectively, according to the manufacturer's instructions. After the separation and quantification of uEVs, 0.5 μg of EVs was applied to each well.

### Statistical analysis

All data are presented as mean ± standard deviation (SD). Two-tailed Student's $t$-test was used to assess the significance of differences between groups. Statistical significance at $P$ values of $< 0.05$ and $< 0.01$ is indicated by $^*P < 0.05$ and $^{**}P < 0.01$, respectively.

## Results

### Study design and variation in uEV protein levels with urine storage conditions

To standardize the sample transfer handling conditions for large-scale validation studies, urine was collected from five patients with BC and stored for 0–6 days under four conditions: with and without pre-processing, at a storage temperature of 4°C, and at RT (20°C) (Fig 1).

Basic information of the five patients who were included in this study is summarized in Table 1. The protein quantities of the EVs separated from each urine sample were analyzed using micro BCA (Fig 2). All five samples were relatively stable with little protein loss when stored at 4°C, regardless of the pre-processing. Precisely speaking, each sample undergoes a pre-processing procedure before EV isolation. However, subsequent to urine collection, our inquiry focused on assessing whether the elimination of bacteria or debris impacts the stability of urinary EVs prior to their isolation. In the case of pre-processed samples stored at RT, almost no protein damage occurred for up to two days. In the case of unprocessed urine samples, the amount of protein decreased after one day at RT. Notably, some samples that were stored at 4°C without pre-processing showed significantly increased protein levels after four days, which may be related to increased sedimentation during storage.

Western blot was performed using uEVs from a representative patient (Patient#3) to investigate changes in the total protein and exosome-specific markers CD63 and CD9, and the same volume of isolated uEVs was loaded in the experiment (Fig 3). The exosome markers CD63 and CD9 were relatively stable in urine stored at 4°C after pre-processing. Under other conditions, the results of the protein amount measurements in Fig 2 were different, and the

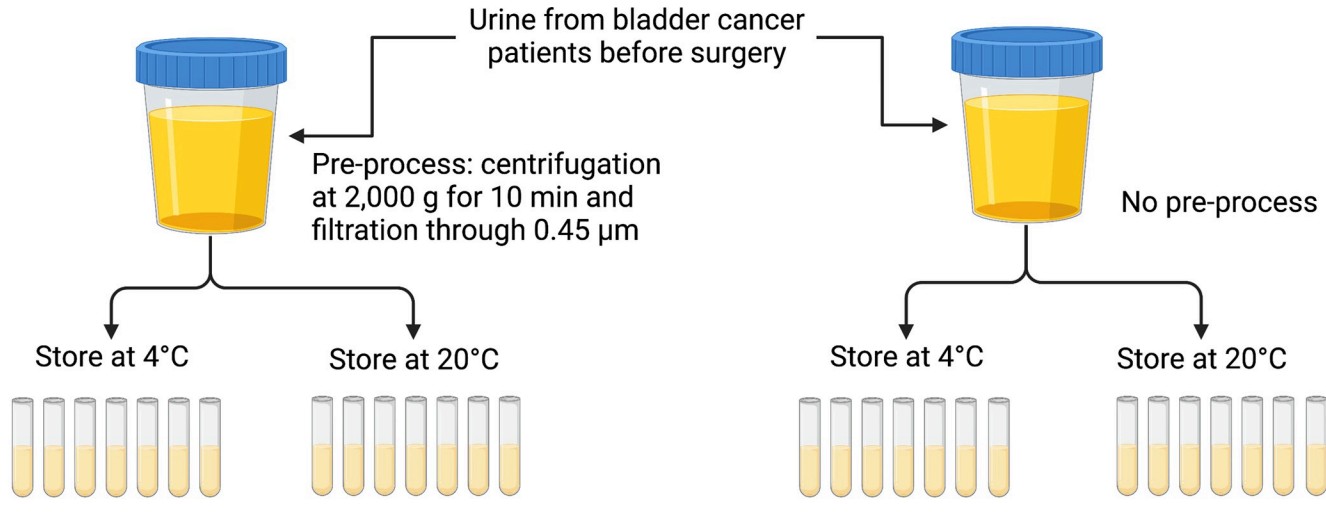

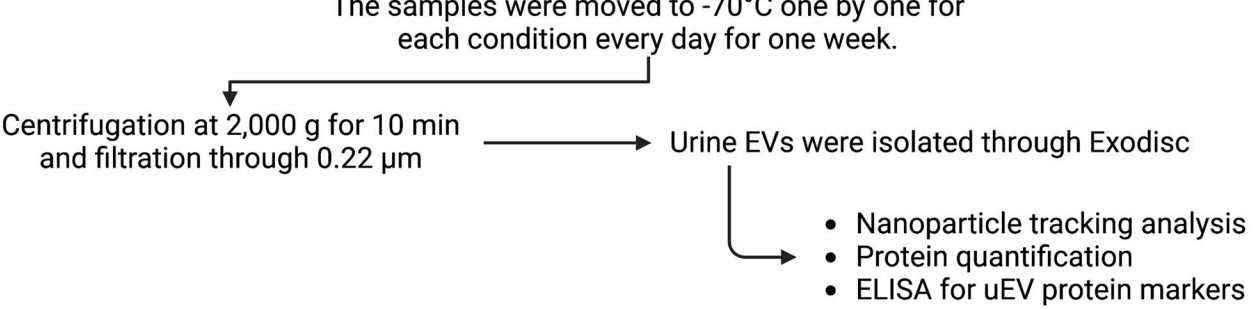

**Fig 1. Schematic diagram of the study design.** The study involved splitting urine samples into two halves, one of which underwent pre-processing, while the other did not. Both halves were then divided into seven vials and stored at either 4˚C or room temperature (20˚C) for six days. One vial from each half was transferred and stored at –70˚C daily for one week. Finally, EVs were isolated using Exodisc and subjected to analysis. Created by Biorender.com.

exosome markers also showed inconsistent expression patterns. While the condition at 20˚C with pre-processing showed a decreased level of CD63 and CD9 proteins over time, no significant decrease was observed in other conditions compared to the 0 d control sample.

**Table 1. Basic information of five patients with bladder cancer.**

|  | Patient #1 | Patient #2 | Patient #3 | Patient #4 | Patient #5 |
|---|---|---|---|---|---|
| Gender / Age | M / 58 | M / 74 | M / 78 | M/53 | M/77 |
| T stage | T1 | T2 | T1 | T2 | T1 |
| Tumor grade | High | High | High | High | High |
| Size (<3cm vs. ≥3cm) | ≥3cm | ≥3cm | ≥3cm | ≥3cm | ≥3cm |
| Multiplicity | Multiple | Multiple | Multiple | Multiple | Multiple |
| Concomitant carcinoma-in-situ | Present | Absent | Present | Absent | Absent |
| Primary vs. Recurrent | Primary | Primary | Primary | Primary | Primary |
| Hematuria, RBC (urine microscopy) | Present (>1/2/HPF) | Present (>1/2/HPF) | Present (>30/HPF) | Present (>1/2/HPF) | Present (>1/2/HPF) |
| Pyuria, WBC (urine microscopy) | Absent | Present (5~9/HPF) | Absent | Absent | Present (10~29/HPF) |

Abbreviations: RBC, red blood cell; WBC, White blood cell; HPF: high power field

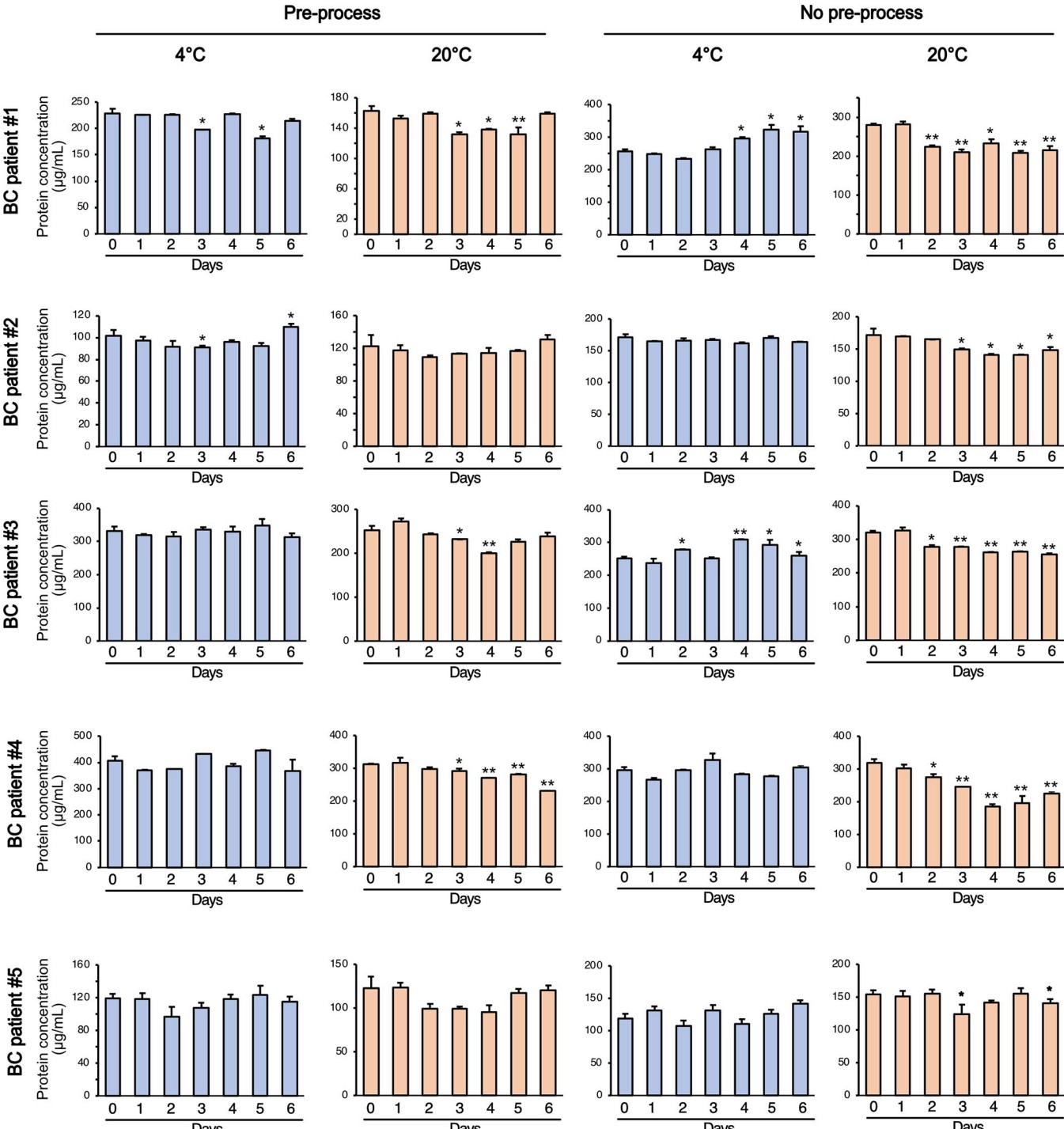

**Fig 2. Analysis of the protein concentration of uEVs from urine samples stored under different conditions.** The uEVs were isolated from each stored urine sample, and their protein concentrations were measured using a micro bicinchoninic acid assay. Data are shown as the mean ± SD, n = 3, *$P < 0.05$, **$P < 0.01$.

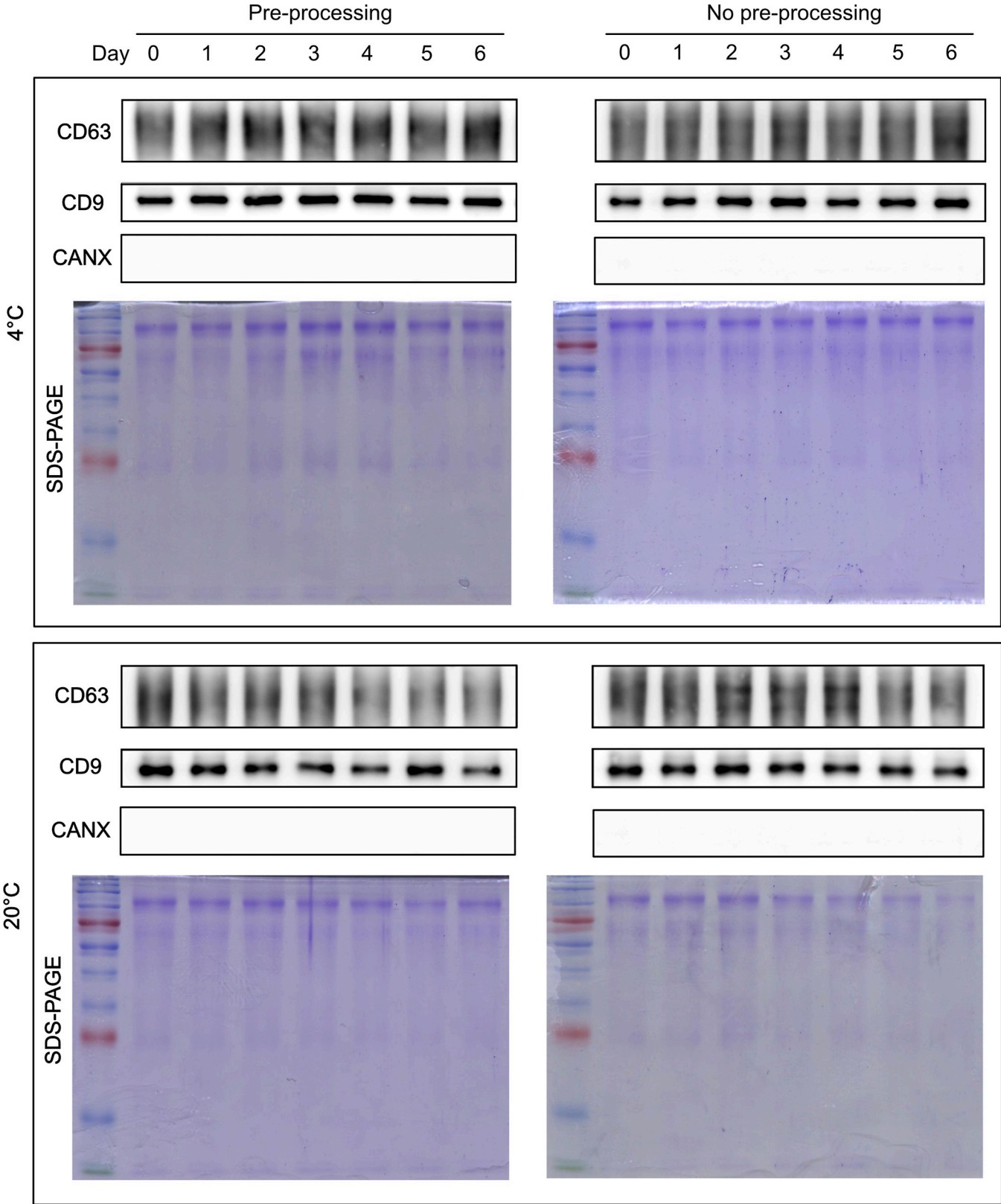

**Fig 3. Western blot analysis for protein markers of uEVs.** CD63, CD9 and calnexin of uEVs stored under different conditions were analyzed using western blotting. Total proteins from each uEV sample were presented by staining on SDS-PAGE gels.

## Effect of urine storage conditions on the number of uEV particles

To determine the number of EV particles in each storage condition, isolated EVs were analyzed using a nanoparticle-tracking analyzer (Fig 4). No evidence suggested that the number of EVs was significantly affected by the pre-processing or storage time. However, when no pre-

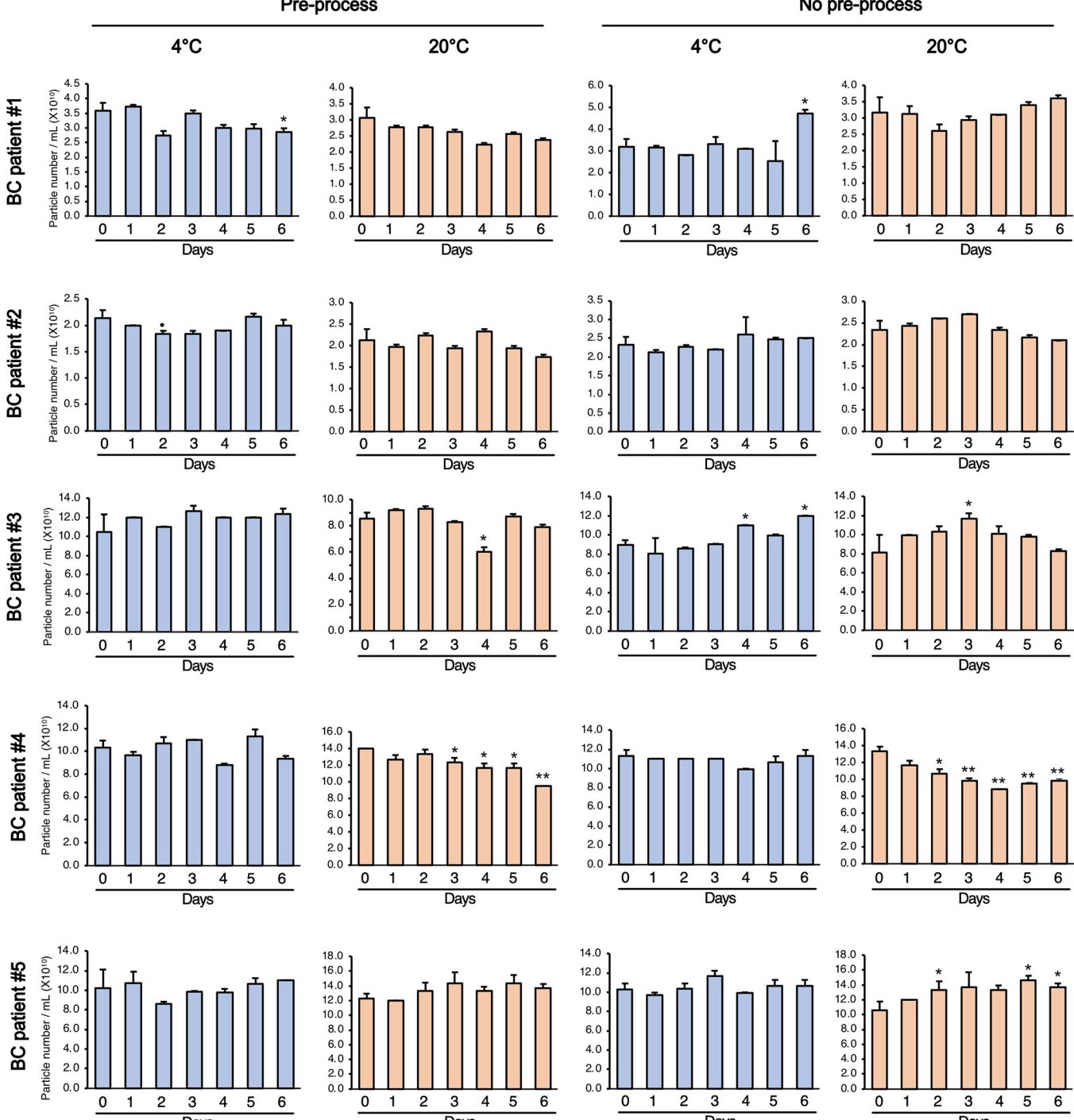

**Fig 4. Analysis of the particle numbers of uEVs from urine samples stored at different conditions.** Using ZetaView analysis, the number of EVs was analyzed from each stored aliquot. Data are shown as mean ± SD, n = 3, *$P < 0.05$.

processing was applied, some EV fractions showed a significant increase in particle number with storage time, which could be due to an increase in protein concentration with increasing precipitation in untreated urine (Fig 2). The mean size and size distribution of uEVs remained largely unchanged across various storage conditions and durations (S1 and S2 Figs).

### A2M and CLU levels in uEVs are differentially affected by the conditions used to store the urine

Since there were some different protein quantities of the isolated uEVs among samples, the same quantity of each uEV was applied to the ELISA for A2M and CLU for normalization (0.5 μg/well, Figs 5 and 6). In A2M ELISA, overall A2M levels were stable for up to three days when stored at 4˚C (Fig 5). However, the stability of A2M exhibited differences between pre-processed and unprocessed samples when stored at RT. In the case of patient #1 and #3, the pre-processed samples demonstrated a notable decline in A2M levels over time at RT. However, when the samples were not pre-processed, the A2M levels remained relatively stable. Conversely, the unprocessed samples from patient #4 showed a decrease in A2M levels after three days of storage at RT. Taken together, pre-processing step does not have a beneficial effect on maintaining the stability of A2M in uEVs.

CLU levels remained stable at 4˚C irrespective of pre-processing status. However, storage at RT resulted in a significant decrease in CLU levels. In particular, uEVs from patients #1 and #3 stored at RT showed a significant reduction in CLU levels (Fig 6). Compared with A2M, CLU was more susceptible to decreasing levels over time when stored at RT. In the pre-processed urine samples from patients #1 and #3, CLU levels significantly decreased after two days of storage at RT. Similarly, in the absence of pre-processing, CLU levels decreased after 1–2 days of storage at RT. To investigate wether there were any morphological changes in uEVs, cryo-transmission electron microscopy (Cryo-TEM) was employed to examine EVs derived from patient #1 and #3. However, there were no notable differences observed between the samples collected on day 0 and day 6 (S3 Fig), suggesting decreased protein level may not be caused by degradation of uEVs. In summary, the results of A2M and CLU ELISA demonstrated that both were stable for up to three days at 4˚C and one day at RT. However, the stability of each protein marker in uEVs varies with temperature, indicating the need to analyze the stability of a specific target protein marker before establishing transport conditions.

## Discussion

Urine is one of the most promising sources of biomarkers because it can be noninvasively obtained in large volumes. Urine also contains uEVs with increasing diagnostic potential for various urinary diseases, including cancer [23], infection [24], and metabolic disorders such as diabetes [25]. However, an ideal methodology for urine collection, storage, and uEV isolation has not been established. Thus, the application of urine to uEV experiments is much more challenging than that of blood. Additionally, urine samples can be easily contaminated, and the analysis can be affected by certain medications or dietary factors, leading to inaccurate results [26, 27]. Therefore, urine specimens must be collected, transported, processed, and stored in a standardized manner.

This research is centered around the critical factor of sample transportation conditions, a key consideration in multicenter studies. Our findings demonstrate that the uEV-derived protein markers A2M and CLU, which hold potential for bladder cancer detection, remain stable in most patients' uEVs when stored at 4˚C for up to 6 days. Notably, only the samples from one patient demonstrated an impact after 4 days, and another patient's samples were affected after three days. The primary aim of this study isn't to establish statistical significance, but

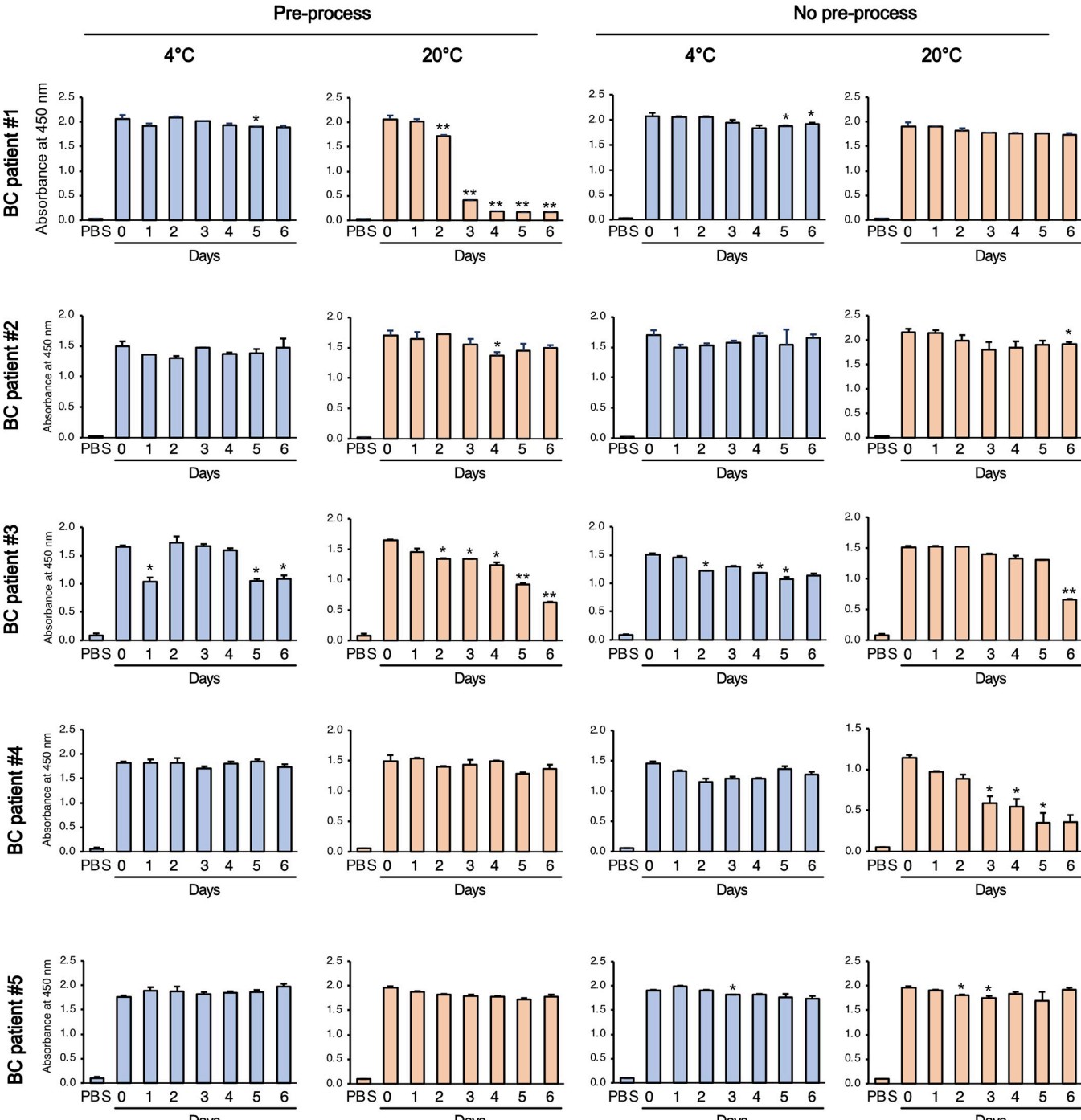

**Fig 5. Analysis of A2M in uEVs from urine samples stored at different conditions.** Equal amounts (0.5 µg/well) of isolated uEVs from urine samples stored under different conditions were used in an A2M ELISA. The resulting absorbance values at 450 nm were presented. Data are shown as mean ± SD, n = 2, *$P < 0.05$, **$P < 0.01$.

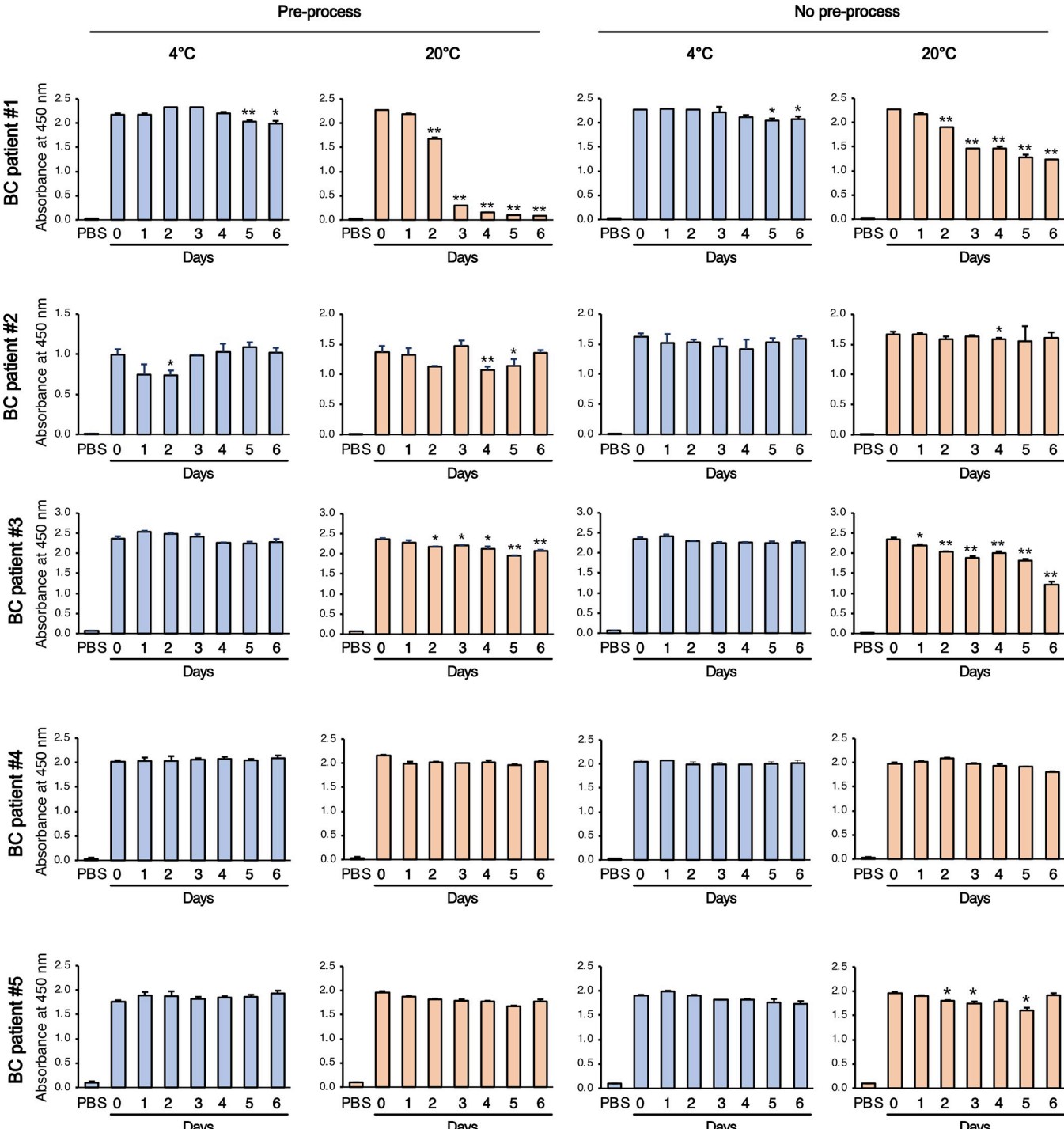

**Fig 6. Analysis of CLU in uEVs from urine samples stored at different conditions.** Equal amounts (0.5 μg/well) of isolated uEVs from urine samples stored under different conditions were used in an CLU ELISA. The resulting absorbance values at 450 nm were presented. Data are shown as mean ± SD, n = 2, *$P < 0.05$, **$P < 0.01$.

rather to identify conditions that ensure the universal preservation of all samples. Our focus was on excluding temperature and time variables that might influence uEV protein markers. This led us to conclude that these markers exhibit stability for up to three days at 4°C without necessitating pre-processing steps. While we acknowledge that drawing robust conclusions from five patients might be constrained, our method entailed an in-depth examination of 28 uEV samples per patient across 7 time points and two distinct temperature conditions, with and without pre-processing. This comprehensive approach resulted in the analysis of a total of 140 uEV samples.

According to the recommendations of the Urine Task Force of the International Society for Extracellular Vesicles regarding sample transportation [5], urine may be aliquoted and transported frozen at –80°C, and for non-aliquoted fresh urine (e.g. home-testing), immediately transfer at RT or 4°C can be considered. The recommended storage temperature for fresh urine samples is at least 4°C and –70°C or cooler for the pre-processed samples [5]. However, prior pre-processing and maintaining –80°C throughout transport requires the use of deep freezers, a constant supply of dry ice, and dedicated laboratory personnel. Additionally, immediate transfer at RT or 4°C is often not feasible for the multicenter sample collection where transport time can vary widely. This can lead to an increased financial burden in uEV research and, subsequently, decreased diagnostic efficiency for uEV protein markers. While general guidelines are stated above, our study results provide a modified transport protocol that is not only more cost-effective for larger cohort studies but also individualized for our protein target of interest.

To our knowledge, this is the first study to examine the effect of different storage temperatures and pre-processing procedures on uEV protein content in urine samples that have not yet undergone EV isolation. There is limited research on the storage conditions of urine samples prior to EV isolation. With the rapid development of EV research, reducing the variability in biomarker discovery and validation during pre-analytical decisions will become increasingly important and more thorough. Furthermore, we tested our hypothesis that syringe filtering as part of pre-processing may be beneficial for ensuring high-quality EV isolation after sample transportation. The current consensus for urine sample processing involves prompt (within 4–6 hours) cooling of fresh urine prior to any processing and urine centrifugation at 800 ×$g$ [5]. We added the filtering step to our pre-processing, making the entire procedure a two-step process of centrifugation at 2,000 ×$g$ for 10 min and filtering through a 0.45 μm syringe. This two-step pre-processing was designed specifically for transportation to further eliminate cells, debris, and bacterial contamination. Contrary to our expectations, syringe filtration did not significantly benefit either uEV-derived A2M or CLU protein yield. Samples without pre-processing stored at 4°C showed similar results as those with pre-processing. At RT, unprocessed A2M showed a decrease in protein quantity from day 5, whereas the pre-processed samples showed a decrease from day 2 (Fig 5). Although the precise underlying factor influencing the stability of A2M or CLU on EVs remains unknown, it is plausible that these proteins could be influenced by pre-processing or temperature variations, given their presence as surface proteins on EVs. the identification of these protein markers using ELISA appeared to be minimally impacted by the lysis step [18]. Hence, we hypothesize that these proteins reside on the surface of urinary EVs. Nevertheless, these results suggest that urine samples for uEV-derived A2M and CLU could be transported from a collection site at 4°C without pre-processing. Our finding that the stability of the two uEV proteins varied under different storage conditions indicates that the analyses of pre-analytical variables on individual uEV biomarker outcomes are mandatory before further validation studies in large samples. To sum up, when transporting urine samples for both A2M and CLU, cooling with ice to maintain 4°C will suffice for up to three days, and no other prompt pre-processing will be necessary.

Although the suggested best practice involves prompt cooling of the sample at collection time, it is unlikely that this universal pre-analytical procedure can be adopted for all uEV studies. Our results suggest that A2M and CLU may safely be stored at RT for up to a day without further protein loss; however, cooling at 4°C will be required if the transportation period becomes longer than that. Taken together, we demonstrated the importance of pre-analytical variables for individual uEV biomarker outcomes as a prerequisite for successful validation studies in large samples.

## Supporting information

**S1 Fig. Mean size of uEVs.** Patient urine samples were collected between day 0 and day 6 of storage. uEVs were separated using Exodisc and analyzed their sizes were assessed using nanoparticle tracking analyzer, Zetaview.
(PDF)

**S2 Fig. Size distributions of uEVs.** A representative urine samples (patient#2) were collected across a span from day 0 to day 6 of storage. The subsequent step involved isolating uEVs using Exodisc, followed by subjecting them to analysis with the nanoparticle tracking analyzer, Zetaview.
(PDF)

**S3 Fig. Cryo-TEM analysis for EVs.** Cryo-TEM was utilized to analyze uEVs obtained from patient #1 and #3. The samples were collected on day 0 and day 6 of storage, and subsequently subjected to cryo-TEM analysis. Red arrows indicate uEVs.
(PDF)

**S1 Dataset.**
(XLSX)

**S1 Raw images.**
(PDF)

## Acknowledgments

We thank the members of Lee's laboratory for technical assistance and helpful discussions.

## Author Contributions

**Conceptualization:** Myung-Shin Lee, Jinsung Park.

**Data curation:** Jisu Lee, Myung-Shin Lee, Jinsung Park.

**Formal analysis:** Jisu Lee.

**Funding acquisition:** Myung-Shin Lee, Jinsung Park.

**Investigation:** Eunha Kim, Joohee Park, Seokjoo Choi, Jinsung Park.

**Methodology:** Jisu Lee, Eunha Kim, Seokjoo Choi, Myung-Shin Lee.

**Resources:** Jinsung Park.

**Supervision:** Myung-Shin Lee, Jinsung Park.

**Visualization:** Myung-Shin Lee.

**Writing – original draft:** Jisu Lee, Joohee Park.

**Writing – review & editing:** Myung-Shin Lee, Jinsung Park.

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
