## [Decision Letter · Decision Letter 0]

1 Jun 2023

PONE-D-23-12716Pre-analytical handling conditions and protein marker recovery from urine extracellular vesicles for bladder cancer diagnosisPLOS ONE

Dear Dr. Lee,

Thank you for submitting your manuscript to PLOS ONE. After careful consideration, we feel that it has merit but does not fully meet PLOS ONE’s publication criteria as it currently stands. Therefore, we invite you to submit a revised version of the manuscript that addresses the points raised during the review process.

We look forward to receiving your revised manuscript.

Kind regards,

Elingarami Sauli, PhD

Academic Editor

PLOS ONE

Journal Requirements:

   "This work was supported by the National Research Foundation of Korea (NRF) grant funded by the Korean government (No. NRF-2022R1F1A1065578 to J.P. and NRF-2022R1I1A3065538 to M-S.L.).NO"

Additional Editor Comments:

You need to increase the number of test cases when validating your protocol, and this should also be compared with proper number of controls. 

Reviewers' comments:

Reviewer's Responses to Questions

**Comments to the Author**

1. Is the manuscript technically sound, and do the data support the conclusions?

Reviewer #1: Partly

Reviewer #2: Yes

Reviewer #3: No

2. Has the statistical analysis been performed appropriately and rigorously? 

Reviewer #1: No

Reviewer #2: Yes

Reviewer #3: N/A

3. Have the authors made all data underlying the findings in their manuscript fully available?

Reviewer #1: Yes

Reviewer #2: Yes

Reviewer #3: Yes

4. Is the manuscript presented in an intelligible fashion and written in standard English?

Reviewer #1: Yes

Reviewer #2: Yes

Reviewer #3: Yes

5. Review Comments to the Author

Reviewer #1: In this study, the authors analyzed the quality of urinary extracellular vesicles (uEVs) stored at 20℃ or 4℃ for 0-6 days. The authors concluded that uEVs were stable when stored at 4℃ for up to 3 days. In this study, only urine samples from three bladder cancer patients were tested. It seems that the number of patients is too small to draw conclusions. For example, the amounts of A2M and CLU were different in the three patients, as shown in Figure 5 and 6. Furthermore, the effects of hematuria or pyuria are unknown. The quality of uEVs in the healthy controls is also unknown.

Reviewer #2: Review Comments

The manuscript titled "Pre-Analytical Handling Conditions and Protein Marker Recovery from Urine Extracellular Vesicles for Bladder Cancer Diagnosis" explores the optimal handling conditions required to maintain the stability of uEV protein markers, alpha-2-macroglobulin (A2M) and clusterin (CLU), in urine sample during transportation.

The authors discovered that storing urine at 4 °C for a maximum of three days and at 20 °C for up to 24 hours resulted in relatively stable levels of A2M and CLU, with minimal impact on analysis results. Interestingly, pre-processing did not significantly improve the preservation of protein markers in uEVs during urine storage.

This study addresses a critical issue in clinical research, as it aims to optimize the clinical protocols utilizing uEVs for diagnostic purposes. Overall, the manuscript is well-written and presents the results in an easily understandable manner.

However, there are a few points that require clarification before publication. These details are outlined below.

- Major points

1. In Fig 2, protein concentration seems to be decreased in pre-processing group compared to the non-processed group.

Was the analysis comparing the pre-processed group to the non-processed group conducted using an equal volume of isolated samples?

2. In Figure 3, the authors described that a Western blot was conducted using uEVs from a representative patient to examine alterations in total protein levels and specific exosome markers CD63 and CD9. The same isolated volume of uEVs was loaded for this experiment. However, Figure 2 indicates that protein concentrations decrease over time in non-processed urine stored at room temperature. Surprisingly, the Western blot results for CD9 and CD63 in all samples from non-processed urine stored at room temperature did not exhibit a significant decrease. How can this discrepancy be explained?

- Minor points

1. For fig 3, the size of CD63 should be specified, and it is recommended to provide complete gel images indicating the target protein size.

2. P6, P9, P10, P12, P13P14, P15, P16 Preprocessing were being used interchangeably, but they should be unified into one.

3. P15, line 8 immediate → immediately

Reviewer #3: In the Ms Lee and collaborators investigated the diagnostic value of alpha-2- 21 macroglobulin (A2M) and clusterin (CLU) in uEVs in respect to storage.

Concerns

TEM of uEV after different storage must be shown.

If they want to validate the protocol, more patients must be recruited

6. PLOS authors have the option to publish the peer review history of their article (what does this mean?). If published, this will include your full peer review and any attached files.

Reviewer #1: No

Reviewer #2: No

Reviewer #3: No

---

## [Author Response · Author response to Decision Letter 0]

13 Jul 2023

Reviewer #1: In this study, the authors analyzed the quality of urinary extracellular vesicles (uEVs) stored at 20℃ or 4℃ for 0-6 days. The authors concluded that uEVs were stable when stored at 4℃ for up to 3 days. In this study, only urine samples from three bladder cancer patients were tested. It seems that the number of patients is too small to draw conclusions. For example, the amounts of A2M and CLU were different in the three patients, as shown in Figure 5 and 6. Furthermore, the effects of hematuria or pyuria are unknown. The quality of uEVs in the healthy controls is also unknown.

Authors’ response: We thank the Reviewer for the excellent suggestions. We agree that number of urine samples is not sufficient to draw a conclusion. However, it was a challenging task to prepare and analyze 42 exosome samples from three patients over a span of six days, all simultaneously, in order to minimize experimental variations. Even for an expert, conducting such experiments is not a trivial matter. Nonetheless, we were able to include two additional urine samples during the revision period, which we believe will enhance the strength of our findings. Additionally, we have added information for the information of pyruia or hematuria (Table 1). In this study, we did not use the healthy control because most uEVs from the healthy controls did not express A2M in our previous study (Lee J, Park HS, Han SR, Kang YH, Mun JY, Shin DW, Oh HW, Cho YK, Lee MS, Park J. Alpha-2-macroglobulin as a novel diagnostic biomarker for human bladder cancer in urinary extracellular vesicles. Front Oncol. 2022 Sep 13;12:976407. doi: 10.3389/fonc.2022.976407. PMID: 36176383; PMCID: PMC9513419.). 

Reviewer #2: Review Comments

The manuscript titled "Pre-Analytical Handling Conditions and Protein Marker Recovery from Urine Extracellular Vesicles for Bladder Cancer Diagnosis" explores the optimal handling conditions required to maintain the stability of uEV protein markers, alpha-2-macroglobulin (A2M) and clusterin (CLU), in urine sample during transportation.

The authors discovered that storing urine at 4 °C for a maximum of three days and at 20 °C for up to 24 hours resulted in relatively stable levels of A2M and CLU, with minimal impact on analysis results. Interestingly, pre-processing did not significantly improve the preservation of protein markers in uEVs during urine storage.

This study addresses a critical issue in clinical research, as it aims to optimize the clinical protocols utilizing uEVs for diagnostic purposes. Overall, the manuscript is well-written and presents the results in an easily understandable manner.

However, there are a few points that require clarification before publication. These details are outlined below.

Authors’ response: We thank the Reviewer for the positive comments, encouragement, and support. Following the great suggestions of the Reviewer, we have revised our manuscript. 

- Major points

1. In Fig 2, protein concentration seems to be decreased in pre-processing group compared to the non-processed group.

Was the analysis comparing the pre-processed group to the non-processed group conducted using an equal volume of isolated samples?

Authors’ response: Yes. We utilized urine samples of the same volume. Following pre-processing, 10 ml of urine was preserved in each tube, while 5 ml of urine was employed for uEVs isolation. As you may be aware, certain proteins have the potential to be captured by the filter during the filtration process, leading to a decline in protein concentration within the pre-processed group.

2. In Figure 3, the authors described that a Western blot was conducted using uEVs from a representative patient to examine alterations in total protein levels and specific exosome markers CD63 and CD9. The same isolated volume of uEVs was loaded for this experiment. However, Figure 2 indicates that protein concentrations decrease over time in non-processed urine stored at room temperature. Surprisingly, the Western blot results for CD9 and CD63 in all samples from non-processed urine stored at room temperature did not exhibit a significant decrease. How can this discrepancy be explained?

Authors’ response: We sincerely appreciate the insightful comments provided by the reviewer. Fig 3 in our study corresponded to the samples obtained from patient #3. As highlighted by the reviewer, we observed a decrease in the total protein content of exosomes stored at room temperature (RT). Interestingly, we found that the protein quantity did not show a strong correlation with the levels of CD63 and CD9. Nonetheless, upon comparing CD63/CD9 between day 0 and day 4 in western blot analysis, some differences were observed. However, overall, we did not observe any clear discrepancies in the western blot results, indicating no significant changes. Additionally, to investigate the possibility of morphological alterations in uEVs, we conducted cryo-transmission electron microscopy (cryo-TEM) analysis (S1 Fig). However, we did not observe any substantial morphological changes in the TEM images.

- Minor points

1. For fig 3, the size of CD63 should be specified, and it is recommended to provide complete gel images indicating the target protein size.

Authors’ response: We thank the reviewer for pointing this out. We have added whole gels for western blot in supplementary figures. 

2. P6, P9, P10, P12, P13P14, P15, P16 Preprocessing were being used interchangeably, but they should be unified into one.

Authors’ response: They have unified into “pre-processing”. 

3. P15, line 8 immediate → immediately

Authors’ response: We have corrected it. 

Reviewer #3: In the Ms Lee and collaborators investigated the diagnostic value of alpha-2- 21 macroglobulin (A2M) and clusterin (CLU) in uEVs in respect to storage.

Concerns

TEM of uEV after different storage must be shown.

If they want to validate the protocol, more patients must be recruited

Authors’ response: We express our gratitude to the reviewer for their valuable suggestions. In response, we have incorporated the TEM image of uEV as S1 Fig in the revised manuscript. Furthermore, we have expanded our dataset by including data from two additional patients.

---

## [Decision Letter · Decision Letter 1]

7 Aug 2023

PONE-D-23-12716R1Pre-analytical handling conditions and protein marker recovery from urine extracellular vesicles for bladder cancer diagnosisPLOS ONE

Dear Dr. Lee,

Thank you for submitting your manuscript to PLOS ONE. After careful consideration, we feel that it has merit but does not fully meet PLOS ONE’s publication criteria as it currently stands. Therefore, we invite you to submit a revised version of the manuscript that addresses the points raised during the review process. When responding to the reviewer comments, please make sure to update your discussions as per the improved sample size....

We look forward to receiving your revised manuscript.

Kind regards,

Elingarami Sauli, PhD

Academic Editor

PLOS ONE

Journal Requirements:

Reviewers' comments:

Reviewer's Responses to Questions

**Comments to the Author**

1. If the authors have adequately addressed your comments raised in a previous round of review and you feel that this manuscript is now acceptable for publication, you may indicate that here to bypass the “Comments to the Author” section, enter your conflict of interest statement in the “Confidential to Editor” section, and submit your "Accept" recommendation.

Reviewer #4: (No Response)

Reviewer #5: All comments have been addressed

2. Is the manuscript technically sound, and do the data support the conclusions?

Reviewer #4: No

Reviewer #5: Yes

3. Has the statistical analysis been performed appropriately and rigorously? 

Reviewer #4: Yes

Reviewer #5: I Don't Know

4. Have the authors made all data underlying the findings in their manuscript fully available?

Reviewer #4: Yes

Reviewer #5: Yes

5. Is the manuscript presented in an intelligible fashion and written in standard English?

Reviewer #4: Yes

Reviewer #5: Yes

6. Review Comments to the Author

Reviewer #4: The manuscript by Lee et al. looks at determining optimal conditions for handling and processing of urine EVs from bladder cancer patients. Although there is always a need to optimise practices within the liquid biopsy field the current manuscript does not adequately do this. There are a number of points which need addressing to significantly strengthen the study.

(1) Previous review has identified the need for a larger patient cohort. The authors have gone some way to address this in the current revised manuscript by increasing their cohort from 3 to 5. The major issue is that most of the conclusions are based on 2 out of 5 patients with no real clear picture. The study really needs to provide additional patients for meaningful conclusions to be made.

(2) The authors need to address the fact all EVs preparations, whether pre-processed or not, were ultimately “processed” as they were filtered through a 0.22 uM filter to remove aggregrates and cellular debris.

(3) Further discussion on how A2M and clusterin are likely to be presented on the surface of EVs would be helpful in understanding the influences of processing and storage on their stability.

(4) Nanoparticle tracking analysis captures both number and size distribution of EVs as stated in methods. Only EV number is provided in the results. The inclusion of size distribution could be very meaningful as an additional distinguishing parameter for establishing overall changes in EV properties.

(5) For western blot analysis of EVs it is not clear whether equivalents numbers of EVs were loaded in each case. The authors state in the methods that EVs were quantified based on protein determination but then state that equal volumes of isolated EVs were loaded on a gel. The authors should provide total protein stains of their westerns as a means of establishing whether equivalent EVs were actually loaded. Quantifying CD9 bands may also help here as well assuming there is no variation. CD63 antibody blots are poor and difficult to interpret. Also blotting for ER-resident marker such as calnexin would establish purity of EV preparations. Blots lack negative controls.

(6) Should highlight on EM images the actual EVs.

(7) The manuscript needs to be updated as text frequently refers to 3 patients rather than 5 patients.

Reviewer #5: All the criticisms raised by the reviewers have been addressed by the authors. In this revised form, the quality of the paper has been significantly improved.

7. PLOS authors have the option to publish the peer review history of their article (what does this mean?). If published, this will include your full peer review and any attached files.

Reviewer #4: No

Reviewer #5: No

---

## [Author Response · Author response to Decision Letter 1]

19 Aug 2023

Reviewer #4: The manuscript by Lee et al. looks at determining optimal conditions for handling and processing of urine EVs from bladder cancer patients. Although there is always a need to optimise practices within the liquid biopsy field the current manuscript does not adequately do this. There are a number of points which need addressing to significantly strengthen the study.

(1) Previous review has identified the need for a larger patient cohort. The authors have gone some way to address this in the current revised manuscript by increasing their cohort from 3 to 5. The major issue is that most of the conclusions are based on 2 out of 5 patients with no real clear picture. The study really needs to provide additional patients for meaningful conclusions to be made.

Authors’ response: We highly value the feedback we've received and wish to address a previous reviewer's suggestion. In the previous manuscript, we expanded our patient cohort from 3 to 5 individuals. Although we acknowledge that drawing strong conclusions from just five patients might be limited, it's crucial to understand the complexity of our experimental setup. Our approach involved analyzing 28 urinary extracellular vesicle (uEV) samples per patient across 7 time points and two temperature conditions, with and without preprocessing. This translated to a total of 140 uEV samples (5 patients × 28 samples). In the initial manuscript version, we analyzed 84 samples from 3 patients concurrently to mitigate potential biases. During the revision, we included an additional 56 uEV samples from 2 more patients.

Our core objective is to identify optimal transport conditions that maintain the integrity of uEV protein markers. The results showed that the majority of uEV protein markers remained stable even when stored at 4︒C for up to 6 days. Only one patient's samples were affected after 4 days, and another patient's after three days. Our study doesn't aim to establish statistical significance. Our focus is on defining conditions that universally preserve all samples. We intend to exclude temperatures and durations that could potentially impact the uEV protein markers. Based on our findings, we concluded that these markers remain stable for up to 3 days at 4︒C, without the necessity of pre-processing steps.

We recognize this limitation but believe our study holds significance by providing valuable insights for uEV protein marker research. The preservation of these markers is of great importance in the collection of urine samples, as it offers clinicians the convenience of shipping without the need for dry ice or extensive pre-processing. Below our revised discussion section reflects this perspective.

“This research is centered around the critical factor of sample transportation conditions, a key consideration in multicenter studies. Our findings demonstrate that the uEV-derived protein markers A2M and CLU, which hold potential for bladder cancer detection, remain stable in most patients’ uEVs when stored at 4°C for up to 6 days. Notably, only the samples from one patient demonstrated an impact after 4 days, and another patient's samples were affected after three days. The primary aim of this study isn't to establish statistical significance, but rather to identify conditions that ensure the universal preservation of all samples. Our focus was on excluding temperature and time variables that might influence uEV protein markers. This led us to conclude that these markers exhibit stability for up to three days at 4°C without necessitating pre-processing steps. While we acknowledge that drawing robust conclusions from five patients might be constrained, our method entailed an in-depth examination of 28 uEV samples per patient across 7 time points and two distinct temperature conditions, with and without pre-processing. This comprehensive approach resulted in the analysis of a total of 140 uEV samples.”

(2) The authors need to address the fact all EVs preparations, whether pre-processed or not, were ultimately “processed” as they were filtered through a 0.22 uM filter to remove aggregrates and cellular debris.

Authors’ response: We thank the reviewer for the great suggestions. We have added the description in Results section like below. 

“Precisely speaking, each sample undergoes a pre-processing procedure before EV isolation. However, subsequent to urine collection, our inquiry focused on assessing whether the elimination of bacteria or debris impacts the stability of urinary EVs prior to their isolation.” 

(3) Further discussion on how A2M and clusterin are likely to be presented on the surface of EVs would be helpful in understanding the influences of processing and storage on their stability.

Authors’ response: We thank the reviewer for the excellent comments. We have added a description for it in the discussion section like below. 

“Although the precise underlying factor influencing the stability of A2M or CLU on EVs remains unknown, it is plausible that these proteins could be influenced by pre-processing or temperature variations, given their presence as surface proteins on EVs. the identification of these protein markers using ELISA appeared to be minimally impacted by the lysis step [18]. Hence, we hypothesize that these proteins reside on the surface of urinary EVs.”

18. Lee J, Park HS, Han SR, Kang YH, Mun JY, Shin DW, et al. Alpha-2-macroglobulin as a novel diagnostic biomarker for human bladder cancer in urinary extracellular vesicles. Front Oncol. 2022;12:976407. Epub 20220913. doi: 10.3389/fonc.2022.976407. PubMed PMID: 36176383; PubMed Central PMCID: PMCPMC9513419.

(4) Nanoparticle tracking analysis captures both number and size distribution of EVs as stated in methods. Only EV number is provided in the results. The inclusion of size distribution could be very meaningful as an additional distinguishing parameter for establishing overall changes in EV properties.

Authors’ response: We express our appreciation to the Reviewer for their valuable recommendations. Due to the relatively insignificant disparities in EV size and distribution across different conditions among patients, we did not incorporate these aspects into the initial manuscript. However, in accordance with the Reviewer's advice, we have now introduced the mean size of uEVs under each condition as Supplementary Figure 1. Additionally, we have included a representative size distribution in Supplementary Figure 2.

(5) For western blot analysis of EVs it is not clear whether equivalents numbers of EVs were loaded in each case. The authors state in the methods that EVs were quantified based on protein determination but then state that equal volumes of isolated EVs were loaded on a gel. The authors should provide total protein stains of their westerns as a means of establishing whether equivalent EVs were actually loaded. Quantifying CD9 bands may also help here as well assuming there is no variation. CD63 antibody blots are poor and difficult to interpret. Also blotting for ER-resident marker such as calnexin would establish purity of EV preparations. Blots lack negative controls.

Authors’ response: We express our gratitude to the Reviewer for their valuable insights. Given that the loaded EVs originated from an equivalent volume of EVs extracted from each urine sample, there could indeed be variations in the count of EV particles. This variability in particle numbers among the EVs is evident in Figure 4. While CD63, CD9, and CD81 are all recognized as tetraspanin EV markers, it is important to acknowledge that they can exhibit diverse expression patterns across different EV samples. In line with the Reviewer's suggestion, we have now incorporated SDS-PAGE gel and western blot analyses for calnexin. Notably, our presentation focuses exclusively on uEVs from patient samples, as uEVs from healthy individuals also express EV markers.

(6) Should highlight on EM images the actual EVs.

Authors’ response: We have indicated EVs in the EM images by arrows. 

(7) The manuscript needs to be updated as text frequently refers to 3 patients rather than 5 patients.

Authors’ response: Thank the Reviewer for the kind indications. We have replaced 3 patients to 5 patients in the manuscript. 

Reviewer #5: All the criticisms raised by the reviewers have been addressed by the authors. In this revised form, the quality of the paper has been significantly improved.

Authors’ response: We express our gratitude to the Reviewer for dedicating significant effort to the evaluation of our manuscript.

---

## [Editor Report · Decision Letter 2]

24 Aug 2023

Pre-analytical handling conditions and protein marker recovery from urine extracellular vesicles for bladder cancer diagnosis

PONE-D-23-12716R2

Dear Dr. Myung Shing-Lee,

We’re pleased to inform you that your manuscript has been judged scientifically suitable for publication and will be formally accepted for publication once it meets all outstanding technical requirements.

Kind regards,

Elingarami Sauli, PhD

Academic Editor

PLOS ONE
---

## [Editor Report · Acceptance letter]

29 Aug 2023

PONE-D-23-12716R2 

Pre-analytical handling conditions and protein marker recovery from urine extracellular vesicles for bladder cancer diagnosis 

Dear Dr. Lee:

I'm pleased to inform you that your manuscript has been deemed suitable for publication in PLOS ONE. Congratulations! Your manuscript is now with our production department. 

Kind regards, 

on behalf of

Dr. Elingarami Sauli 

Academic Editor

PLOS ONE